# Lactoferrin Alleviated AFM1-Induced Apoptosis in Intestinal NCM 460 Cells through the Autophagy Pathway

**DOI:** 10.3390/foods11010023

**Published:** 2021-12-23

**Authors:** Hongya Wu, Yanan Gao, Songli Li, Xiaoyu Bao, Jiaqi Wang, Nan Zheng

**Affiliations:** 1Key Laboratory of Quality & Safety Control for Milk and Dairy Products of Ministry of Agriculture and Rural Affairs, Institute of Animal Sciences, Chinese Academy of Agricultural Sciences, Beijing 100193, China; wuhongya0304@163.com (H.W.); gyn758521@126.com (Y.G.); lisongli@caas.cn (S.L.); baoxy_hzau@126.com (X.B.); wangjiaqi@caas.cn (J.W.); 2Laboratory of Quality and Safety Risk Assessment for Dairy Products of Ministry of Agriculture and Rural Affairs, Institute of Animal Sciences, Chinese Academy of Agricultural Sciences, Beijing 100193, China; 3Milk and Dairy Product Inspection Center of Ministry of Agriculture and Rural Affairs, Institute of Animal Sciences, Chinese Academy of Agricultural Sciences, Beijing 100193, China; 4State Key Laboratory of Animal Nutrition, Institute of Animal Sciences, Chinese Academy of Agricultural Sciences, Beijing 100193, China

**Keywords:** lactoferrin, aflatoxin M1, apoptosis, autophagy

## Abstract

Aflatoxin M1 (AFM1) is the only mycotoxin with maximum residue limit in milk, which may result in serious human diseases. On the contrary, lactoferrin (Lf) is an active protein with multiple functions. Studies have confirmed that Lf has a powerful potential to protect the intestines, but the influence of Lf on mycotoxins is not clear. This study aims to explore whether Lf can protect the cytotoxicity induced by AFM1, and determine the underlying mechanisms in human normal colonic epithelial NCM460 cells. The results indicated that AFM1 decreased the cell viability, and increased the levels of apoptosis and autophagy of NCM460 cells. Lf can alleviate the cytotoxicity induced by AFM1 through enhancing cell viability, significantly down-regulated the expression of apoptotic genes and proteins (*BAX*, *caspase3*, *caspase9*, caspase3, and caspase9), and regulated the gene and protein expression of autophagy factors (*Atg5*, *Atg7*, *Atg12*, *Beclin1*, *ULK1*, *ULK2*, LC3, and p62). Furthermore, interference of the key gene Atg5 of autophagy can reduce AFM1-induced apoptosis, which is consistent with the role of Lf, implying that Lf may protect AFM1-induced intestinal injury by inhibiting excessive autophagy-mediated apoptosis. Taken together, our data indicated that Lf has a mitigating effect on apoptosis induced by AFM1 through the autophagy pathway.

## 1. Introduction

Milk provides energy for the human body, and contains a variety of nutrients, but there are also some harmful factors [1]. Among them, aflatoxin M1 (AFM1) is one of the main risks that threaten its quality and safety [2,3,4], and pollution conditions are widespread in countries around the world, such as China, Iran, and Morocco [5,6,7]. The general milk processing methods (heating and fermentation) have little effect on AFM1, which proves that it is very stable [8]. More importantly, AFM1 has been classified as a “Group 1 Human Carcinogen”, which is sufficient to prove its high toxicity [9]. In recent years, the study of AFM1 on intestinal toxicity has become a research hotspot. Research evidences have shown that AFM1 has a damaging effect on intestinal cells, including decreased cell viability [10], cell cycle arrest [11], and genetic damage [12]. However, the study of AFM1 on intestinal cell apoptosis still has great development space.

Lactoferrin (Lf) is a glycoprotein with about 700 amino acids (80 kDa) that can bind to iron [13,14]. The most studied protein in milk is Lf, and its versatile biological activities have been confirmed, including iron metabolism [15], anti-cancer [16], anti-inflammatory [17], antibacterial [18,19], and immunomodulation [20]. There are some studies on the protective effect of Lf on harmful factors. Lf can bind to host cells to protect mice and cells from enterovirus 71 infection [21]. It also plays a protective role against lipopolysaccharide-mediated intestinal mucosal injury and impaired intestinal epithelial barrier function by increasing cell viability and reducing intestinal permeability [22]. In a rat model of carbon tetrachloride administration, it was found that Lf can reduce oxidative stress, and protect liver damage [23]. However, the effect of Lf on the intestinal injury caused by mycotoxins is rarely reported. Only one study demonstrated that Lf can inhibit cytotoxicity and DNA damage caused by mycotoxins [24].

Studies have demonstrated that both AFM1 and Lf are closely related to cell apoptosis, mainly focusing on the JNK and WNT signaling pathway [25,26]. Apoptosis and autophagy play important roles in maintaining the stability of the intracellular environment, which usually occurs when the cell is under stress [27]. The interaction between them is very intricate. Usually, autophagy occurs before apoptosis, and prevents apoptosis [28]. Studies have made clear that increasing autophagy will reduce the cytotoxicity caused by T-2 toxin, and prevent cell apoptosis [29]. However, there is evidence that autophagy may be the executor of inducing cell apoptosis. Excessive activation of autophagy can lead to over self-digestion and degradation of basic cell components, which ultimately leads to cell death [30,31]. The method of limiting autophagy imbalance may be an effective means to protect cells from damage [32]. Therefore, exploring the mechanism of interaction between cell apoptosis and autophagy is perhaps a promising strategy to protect intestinal function.

Intestinal epithelial cells (IECs) are located on the surface of intestinal epithelium, and have an unusual function in immune defense, nutrient absorption, and maintaining body health [33]. IECs are the body’s first barrier against mycotoxins. After ingestion of mycotoxins and other pollutants, the exposure level of IECs was higher than that of other tissues. Based on the previous research foundation, our research aims to investigate the impact of Lf on AFM1-induced human normal IECs NCM460 cells, and further reveal its possible mechanism. This is expected to offer more accurate evidence for the functional study of Lf.

## 2. Materials and Methods

### 2.1. Chemicals and Materials

The AFM1 powder used in this experiment was provided by Pribolab (Qingdao, China). Dissolve AFM1 in dimethyl sulfoxide (DMSO) (Sigma-Aldrich, St. Louis, MO, USA) to prepare 200 μg/mL stock solution, and then dilute it to the required concentration with culture medium. Lf powder was obtained from Sigma-Aldrich (St. Louis, MO, USA). Dissolve Lf power in culture medium to make 10 mg/mL stock solution, and then dilute it to the required concentration. Nonessential amino acids, Dulbecco’s modified Eagle’s medium (DMEM), fetal bovine serum (FBS), and Opti-MEM Medium were purchased from Gibco (Carlsbad, CA, USA). Trypsin (2.5%), antibiotics (100 units/mL penicillin, 100 μg/mL streptomycin), nonessential amino acids (NEAA), Enhanced Cell Counting Kit-8 (CCK-8), and Annexin-V-FITC Staining Kit were provided by Beyotime Biotechnology (Shanghai, China). Trizol Reagent Kit and Lipofectamine 2000 were obtained from Invitrogen (Carlsbad, CA, USA). The Prime Script RT Reagent Kit (Perfect Real Time) and the TB Green Premix Ex Taq II (Tli RNaseH Plus) were obtained from Takara (Shiga Prefecture, Japan). The β-actin and caspase9 antibodies were provided by Bioss (Beijing, China); and the Atg5, SQSTM1/p62, LC3B, caspase3 antibodies, and Atg5-specific siRNAs were provided by Cell Signaling Technology (Danvers, MA, USA).

### 2.2. NCM460 Cell Culture

The NCM460 cells used in this experiment were acquired from INCELL (San Antonio, TX, USA). The cells were grown in DMEM with 10% FBS, 1% antibiotics, and NEAA at 37 °C in a 95% humidified atmosphere with 5% CO_2_, as described previously [34]. A part of NCM460 cells (1 × 10^4^) cultured above was seeded onto 96-well plates. After culturing for 24 h, they were exposed to different concentrations of bovine Lf from 0 to 1000 μg/mL. The other part of cells (1 × 10^5^) was seeded into 6-well plates, and then treated with 8 μg/mL AFM1 with and without Lf for 24 h.

### 2.3. Cytotoxicity Assay

To evaluate the cell survival rate, we used the CCK-8 Kit method. After NCM460 cells grew to about 80–85%, they were incubated with Lf (0, 20, 100, 200, 500, 1000 μg/mL) and together with 8 μg/mL AFM1. Then, we abided by the manufacturer’s guidelines to determine the cell survival rate. Afterward, we used a microplate reader (Thermo Labsystems, Philadelphia, PA, USA) to measure the optical density value at 450 nm.

### 2.4. Annexin-V-FITC Apoptosis Analysis

NCM460 cell samples were collected from each group (control group, Lf group, AFM1 group, Lf and AFM1 combined group). Then, we washed the cells with PBS, and they were resuspended in a Annexin-V-FITC and propidium iodide (PI) solution. Afterwards, they were incubated in the dark at room temperature for 20 min before analysis via flow cytometry, and gated according to Annexin-V-FITC and PI. Necrotic only refers to PI positive cells, whereas early apoptotic only refers to Annexin-V positive cells, and late apoptotic refers to double positive cells.

### 2.5. RT-qPCR Analysis for Determination of Gene Expression

First, we used the Trizol method to extract total RNA from NCM460 cells. Then, the RNA was reverse transcribed into cDNA by a synthesis kit. Finally, the cDNA was amplified by the PCR reaction kit. Amplification was initiated at 95 °C for 60 s, followed by 40 cycles at 95 °C for 10 s, and 60 °C for 30 s. We used the 2−ΔΔCt method to quantify the relevant expression of the target gene, and normalized it to the expression of β-actin.

### 2.6. Western Blotting Analysis for Determination of Protein Expression

The protein expression of β-actin, caspase3, caspase9, p62, and LC3 were detected. First, the same amount of protein was loaded and electrophoresed, and then electroblotted on the polyvinylidene fluoride membrane. After blocking the membrane with 5% blocking buffer (skimmed milk) for 1.5 h, we incubated with the required primary antibody (anti-actin antibody was diluted 1:5000, others were diluted 1:1000) at 4 °C all night. Then, after washing with TBST, we incubated the membrane with the required secondary antibody (The dilution ratio is 1:5000) at room temperature for 1 h. Finally, ImageJ 2× software (Version 2.1.0, National Institutes of Health, Bethesda, MD, USA, 2006) was used to analyze the band densities. The intensity values were normalized to β-actin.

### 2.7. Confocal Microscopy

The change of LC3 positive level was observed using a confocal microscope. After the cells were incubated overnight, they were treated with Lf and AFM1 alone or in combination for 24 h, and washed and fixed with 4% paraformaldehyde fixative, and then blocked at 37 °C. Next, we incubated the cells with the primary antibody overnight and the fluorescent secondary antibody for 1 h. Finally, the number of LC3 punctate positive cells was observed with a confocal microscope.

### 2.8. RNA Interference of Atg5

First, we cultured NCM460 cells (0.25 × 10^6^) in 6-well plates as described above, and then, Atg5 siRNA was diluted in transfection medium (Opti-MEM). Meanwhile, Lipofection 2000 was diluted in Opti-MEM. Subsequently, the diluted two reagents were mixed, and then incubated at room temperature for 5 min. After 48 h of transfection, the cells were exposed to drug treatment for 12 h, and then treated samples were collected for further experiments.

### 2.9. Statistical Analysis

Analysis of data was performed using GraphPad Prism 8.0 (GraphPad software, San Diego, CA, USA) software. A one-way ANOVA test was used, followed by Tukey’s multiple comparison test for statistical analysis of the differences. *p* < 0.05 was considered statistically significant.

## 3. Results

### 3.1. Lf Alleviated AFM1-Induced Cytotoxicity

Cell viability was determined after treatment of Lf (0–1000 μg/mL) (Figure 1A). The result indicated that Lf significantly increased NCM460 cell viability from 100 μg/mL to 1000 μg/mL (*p* < 0.05). Then, cell viability was evaluated after being exposed to single and combination 8 μg/mL AFM1 and different concentrations (0, 20, 100, 200, 500, 1000 μg/mL) of Lf for 48 h. As shown by Figure 1B, 100 μg/mL, 200 μg/mL, 500 μg/mL, and 1000 μg/mL Lf could alleviate survival reduction caused by AFM1. A concentration of 100 μg/mL Lf could significantly enhance the cell survival rate (*p* < 0.05). The combination of 100 μg/mL Lf and AFM1 does not affect the cell survival, and 100 μg/mL is the lowest concentration that does not affect cell viability. Therefore, 100 μg/mL was chosen as the treatment concentration of Lf for subsequent experiments.

### 3.2. Lf Ameliorated AFM1-Induced Apoptosis in NCM460 Cells

As can be seen from Figure 2A,B, 8 μg/mL AFM1 significantly increased the NCM460 cell apoptosis rate (*p* < 0.05). However, 100 μg/mL Lf significantly down-regulated the level of apoptosis induced by AFM1 (*p* < 0.05). Moreover, detection of apoptosis genes found that Lf significantly reduced the mRNA level of *BAX*, *caspase3*, and *caspase9* (*p* < 0.05) (Figure 2C). Subsequently, western blotting results indicated that compared with the AFM1 group, the relative expression of caspase3 and caspase9 in the AFM1 plus Lf group were reduced significantly (*p* < 0.05) (Figure 2D,E). All the above results indicate that Lf alleviated AFM1-induced apoptosis of NCM460 cells.

### 3.3. Lf Alleviated AFM1-Induced Autophagy in NCM460 Cells

To understand whether Lf can reduce AFM1-induced autophagy, autophagy-related genes, proteins, and LC3 positive status were detected and observed (Figure 3). First, our study indicated that the relative mRNA level of autophagy-related genes in the AFM1 group was significantly higher than in the control group (*p* < 0.05) (Figure 3A). In addition, no significant changes could be observed in the Lf treatment group, but the combined treatment of Lf and AFM1 could significantly reduce the mRNA level of autophagy genes (*Atg5*, *Atg7*, *Atg12*, *BECN1*, *ULK1*, and *ULK2*) (*p* < 0.05). The subsequent western blotting results indicated that the expression of p62 decreased after AFM1 treatment, but LC3-II/LC3-I increased. The Lf plus AFM1 group increased the expression of p62, and decreased the expression of LC3-II/LC3-I (Figure 3B,C). Furthermore, confocal microscopy showed that Lf could reduce the positive level of LC3 in NCM460 cells induced by AFM1 (Figure 3D), and the results were the same as those presented by the above genes and proteins. All the above results showed that Lf alleviated AFM1-induced autophagy in NCM460 cells.

### 3.4. Autophagy Interference Alleviated AFM1-Induced Apoptosis and Autophagy in NCM460 Cells

Figure 4A is the effect diagram of interference against Atg5. Then, the apoptosis proteins caspase3 and caspase9, and the autophagy proteins p62 and LC3 were detected after interference (Figure 4B,C). Compared with the non-interfering AFM1 treatment group, the expression of caspase3 and caspase9 were decreased significantly in the interfering AFM1 group (*p* < 0.05), and the relative expression of LC3-II/LC3-I was decreased correspondingly. At the same time, the expression of p62 protein showed a significant increase (*p* < 0.05). It showed that autophagy interference alleviated AFM1-induced apoptosis and autophagy. This is consistent with the role played by Lf mentioned above.

## 4. Discussion

Mycotoxins pollution has had a negative impact on humans and animals, causing serious diseases and a large amount of economic losses, which have been widely concerned around the world [35]. Due to the high frequency of AFM1 in dairy products, it has become a serious problem that has plagued consumers and the whole dairy industry, so it has attracted more and more attention from research [36]. Although the multifunctional activity of Lf has been widely confirmed, research on the related effects of Lf on mycotoxins has rarely been involved. Therefore, in this study, we focused on the mechanism of the protection function of the Lf on AFM1-induced intestinal toxicity in NCM460 cells.

As shown in Figure 5, our study indicates that Lf can enhance cell viability; down-regulate apoptosis-related genes and proteins, and autophagy-related genes and proteins; and up-regulate the relative expression of autophagy protein p62 to alleviate the NCM460 cytotoxicity induced by AFM1. At the same time, we proved that the results of interfering with the key autophagy gene Atg5 are consistent with the results presented by Lf, indicating that Lf may play a protective effect on AFM1-induced intestinal injury by inhibiting apoptosis caused by excessive autophagy.

First, we found that Lf showed a proliferative effect starting from 100 μg/mL (Figure 1A), which is consistent with studies both in vitro mouse crypt cells [37] and Caco-2 cells [38,39]. Moreover, our results showed that Lf had a tendency to increase the decline in cell viability caused by AFM1 (Figure 1B), which was the same as the results of Lf and AFM1 combined in the study of intestinal Caco-2 cells [24], indicating that Lf has a certain protective effect on AFM1-induced cell damage.

Apoptosis can effectively remove damaged cells, and is a form of programmed death, which is mediated by caspase [40]. Responses of various physiological functions, and pathological tissues and organs have shown that the dysregulation of apoptosis may lead to serious intestinal diseases [41,42]. Flow cytometry showed that 100 μg/mL Lf significantly inhibited the amount of apoptosis induced by AFM1 (Figure 2A,B), which is consistent with the previous study results of 100 μg/mL Lf acting on osteoblasts [43]. Moreover, apoptosis is regulated by a variety of apoptotic factors, among which, caspase is the key mediator of programmed cell death. When cells are stimulated by apoptosis, caspase9 is activated, and then caspase9 cleaves and activates caspase3, resulting in apoptosis [44,45]. We found that Lf can down-regulate the expression of intestinal genes caspase9, caspase3, and their corresponding target proteins in NCM460 cells (Figure 2C–E). A recent study showed that Lf can reduce hydrogen peroxide-induced apoptosis in bone mesenchymal stem cells through inhibiting the activation of apoptotic proteins, and another study found that Lf can inhibit dexamethasone-induced osteoarthritis chondrocyte damage [46,47]. These results confirmed that Lf could affect the apoptosis of NCM460 cells induced by AFM1, and play a potential anti-apoptotic role by regulating the expression of apoptotic genes and proteins.

Autophagy is regulated by autophagy-related genes and proteins [48,49]. Atg5 is an essential member of the autophagy process, and plays a non-negligible role in phagophore expansion and autophagosome formation [50]. ULK1 and ULK2 participate in autophagy induction, and ULK1 has a central function [51,52]. P62 and LC3 proteins are widely used to detect the occurrence of autophagy [53], in which p62 level is negatively correlated with autophagy level. Soluble LC3-I type is transformed into lipid bound LC3-II type, and its proportion determines autophagy. We confirmed that Lf may inhibit the excessive autophagy of NCM460 cells induced by AFM1, and reduce the cell damage by down-regulating the level of the autophagy gene and the expression of LC3-II/LC3-I, while up-regulating the relative expression of p62 proteins (Figure 3).

There are extensive interactions between apoptosis and autophagy. Previously, studies have pointed out that basic autophagy has a protective effect on poison-induced cells, but abnormal and excessive autophagy can lead to apoptosis, and aggravate toxicity [54]. Interference against the Atg gene is usually used as an effective means to regulate autophagy. In our study, we found that siRNA-Atg5 transfection could inhibit autophagosome formation, and reduce the expression of AFM1-induced apoptosis proteins and autophagy proteins (Figure 4). This is consistent with the findings of emulsified isoflurane acting on fetal neural stem cells. [55]. All suggest that the appropriate level of autophagy can be maintained through regulating the Atg5 autophagy pathway. Further, both Lf and siRNA-Atg5 treatment can reduce AFM1-induced apoptosis, which means that Lf may protect AFM1-induced intestinal injury by inhibiting excessive autophagy-mediated apoptosis.

## 5. Conclusions

In conclusion, Lf can effectively protect NCM460 cells from AFM1-induced apoptosis through the autophagy pathway. Therefore, this shows that Lf can be used as a strong anti-apoptotic substance to resist the toxicity of AFM1. It is confirmed that the active substances in milk (such as Lf) have a protective effect on the damage caused by harmful factors in milk (such as AFM1), which is helpful to explore the “nutrition safety” balance of milk itself, expand people’s understanding of food safety, and improve consumers’ confidence in food safety.

## Figures and Tables

**Figure 1 foods-11-00023-f001:**
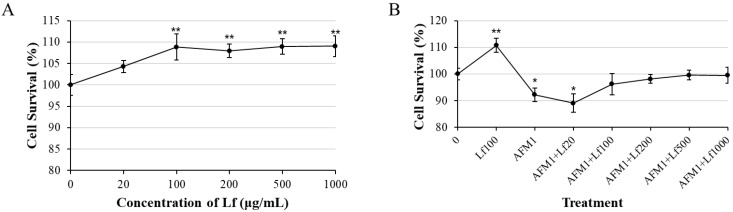
The effect of Lf on the proliferation of NCM460 cells. We examined the proliferation effects after NCM460 cells incubated with Lf (**A**) and together with 8 μg/mL AFM1 (**B**) for 24 h. Results represented as means ± SD. * *p* < 0.05, ** *p* < 0.01 vs. control.

**Figure 2 foods-11-00023-f002:**
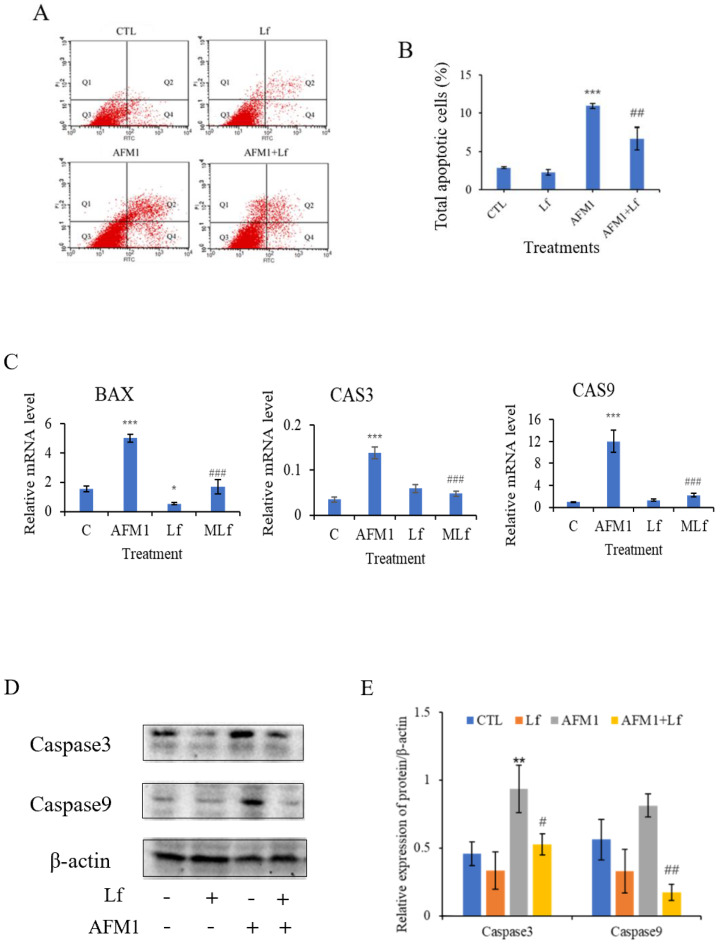
Lf ameliorated AFM1-induced apoptosis in NCM460 cells. The apoptosis rate of cells exposed to 8 μg/mL AFM1 with or without Lf of 100 μg/mL. (**A**) Flow cytometry plots of NCM460 cells. (**B**) The sum of the early and late apoptotic cell populations was the total apoptotic cells. (**C**) Relative mRNA levels of apoptosis genes. C represents control, MLf represents combined AFM1 and Lf. (**D**,**E**) Immunoblotting was performed on protein extracts for autophagy proteins and band densities quantification. Results represented as means ± SD. * *p* < 0.05, ** *p* < 0.01, *** *p* < 0.001 vs. control (C/CTL). # *p* < 0.05, ## *p* < 0.01, ### *p* < 0.001 vs. AFM1.

**Figure 3 foods-11-00023-f003:**
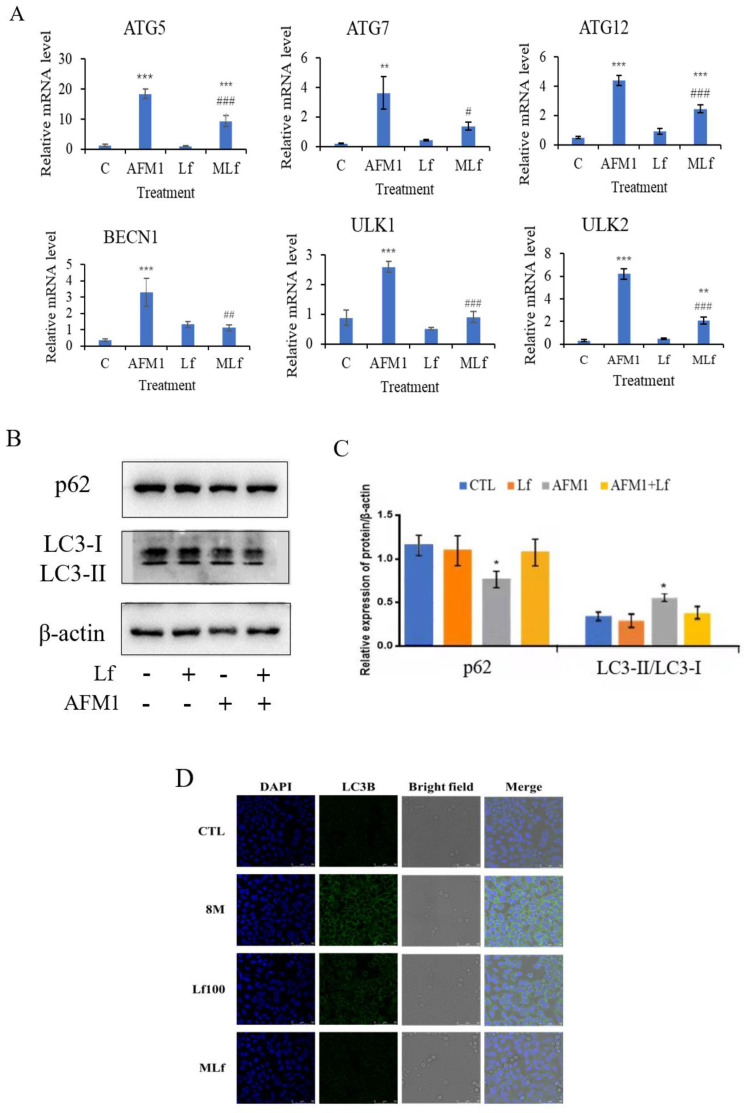
Lf alleviates AFM1-induced autophagy in NCM460 cells. 100 μg/mL Lf and 8 μg/mL AFM1 were treated in NCM460 cells alone or in combination. (**A**) Relative mRNA levels of autophagy genes. (**B**,**C**) Immunoblotting was performed on protein extracts for autophagy proteins and band densities quantification. (**D**)The changes of LC3 positive level were analyzed by confocal microscope. Results represented as means ± SD. * *p* < 0.05, ** *p* < 0.01, *** *p* < 0.001 vs. control (C/CTL). # *p* < 0.05, ## *p* < 0.01, ### *p* < 0.001 vs. AFM1.

**Figure 4 foods-11-00023-f004:**
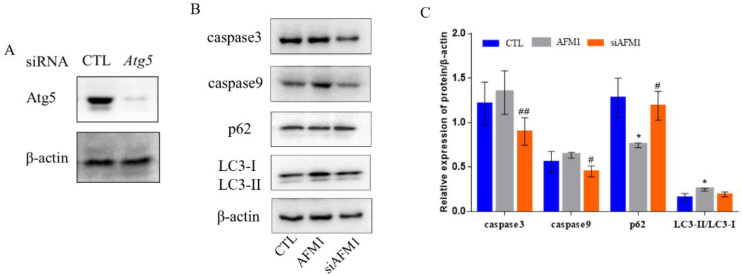
The apoptosis and autophagy of NCM460 cells transfected with siRNA-Atg5. (**A**) Transfection effect of siRNA-Atg5 in NCM460 cells. (**B**,**C**) Relative expression of apoptosis and autophagy proteins with or without transfected NCM460 cells after treatment with AFM1 for 12 h. Results represented as means ± SD. * *p* < 0.05 vs. control (CTL). # *p* < 0.05, ## *p* < 0.01 vs. AFM1.

**Figure 5 foods-11-00023-f005:**
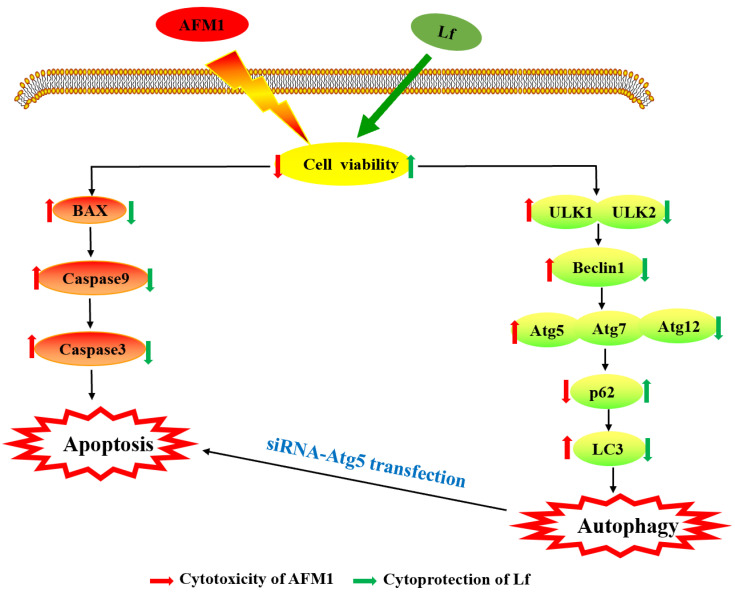
Summary diagram showing the cytotoxicity induced by AFM1, and the cytoprotective effect of Lf in NCM460 cells.

## Data Availability

The datasets generated for this study are available on request to the corresponding author.

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
