# Peer review of "Lactoferrin Alleviated AFM1-Induced Apoptosis in Intestinal NCM 460 Cells through the Autophagy Pathway"

_foods, 2021, doi:10.3390/foods11010023_

Round 1

Reviewer 1 Report

The authors presented a study that aimed to verify the ability of Lactoferrin to protect AFM1-induced cytotoxicity and determine the underlying mechanisms in NCM460 cells.

The methodology used is described in detail and is appropriate to support the hypothesis. Results and discussion are consistent, present in a well-structured manner, and totally relevant to the field

Author Response

Thanks for the reviewer's positive comments. We will continue to work hard.

Reviewer 2 Report

The interest of this study is to observe the impact of Lf on normal human NCM460 IEC cells induced by AFM1. Which is relevant because AFM1 is found in a food of high human consumption such as milk. The study is presented in a clear and simple way. Just a few of the following minor formatting fixes:

Line 38 separate the words [9].In recent

Line 97 CO2 put 2 as subscript

Line 98 (1 × 104) put 4 as the supra index

Line 99 (1 × 105) put 5 as supra index

Line 140 (0.25 × 106) put 6 as supra index

Line 168 separate the words Fig.2A, like Fig. 2A

Line 209-211 There is an error in the figure because it is mentioned as Fig. 5, but it should be Fig. 4

Line 168 AFM1.It separates the words

In the references section, all of them have typing errors due to words pasted, it is suggested to correct them.

Reviewer 3 Report

In subjected paper, the Authors presented studies upon the investigation the impact of lactoferrin on aflatoxin M1 induced human normal IECs NCM460 cells, and further reveal its possible mechanism.

The topic undertaken by the authors is interesting. Authors presented this issue in a clear way by describing it correctly in a chapter “Materials and Methods”. They also presented and discussed those results in chapters “Results” and “Discussion”.

Author Response

(The authors gave the same response as above.)
